# FreeMorph: Tuning-Free Generalized Image Morphing with Diffusion Model

## Abstract

We present **FreeMorph**, the first tuning-free method for image morphing that accommodates inputs with varying semantics or layouts. Unlike existing methods, which rely on fine-tuning pre-trained diffusion models and are limited by time constraints and semantic/layout discrepancies, FreeMorph delivers high-fidelity image morphing without extensive training. Despite its efficiency and potential, tuning-free methods still face challenges in maintaining high-quality image morphing due to the non-linear nature of the multi-step denoising process and bias inherited from the pre-trained diffusion model. In this paper, we introduce FreeMorph to address this challenge by integrating two key innovations. **1)** We first propose a **guidance-aware spherical interpolation** design that incorporates the explicit guidance from the input images by modifying the self-attention modules, addressing identity loss, and ensuring consistent transitions throughout the generated sequences. **2)** We further introduce a **step-oriented motion flow** that blends self-attention modules derived from each input image to achieve controlled and directional transitions that respect both input images. Our extensive evaluations demonstrate that FreeMorph outperforms existing methods with training that is $10\times \sim 50\times$ faster, establishing a new state-of-the-art for image morphing. The code will be released.

## 1 Introduction

Given two distinct input images, image morphing (Zope & Zope, 2017; Nri, 2022) aims to gradually change the attributes such as shape, texture, and overall image layout to produce a series of intermediate images that transit smoothly from one image to the other. This process is now widely utilized in fields such as animation, film transitions, and photo-editing tools (Aloraibi, 2023; Wolberg, 1996; 1998), offering an effective means to enhance creative expression and imagination. Historically, image morphing has relied on image warping (Smythe, 1990; Wolberg, 1990; Fant, 1986) for aligning corresponding points and color interpolation (Beier & Neely, 1992; Lee et al., 1998) for blending. These methods, however, often fall short in handling complex textural and semantic transitions, making them less effective for images with intricate details. With the advancements in deep learning, GANs (Goodfellow et al., 2014; Karras et al., 2019; Brock et al., 2019; Sauer et al., 2023) and VAEs (Kingma & Welling, 2013) have significantly improved image morphing by allowing latent code interpolations. Despite their capabilities, these approaches still face challenges with real-world images due to limited training data and issues with information loss during GAN inversion. This underscores the need for more identity-preserved and generalized methods in image morphing.

Recently, thanks to the collections of large datasets with extensive text-image pairs, vision-language models (*e.g.*, Chameleon (Team, 2024)), diffusion models (*e.g.*, Stable Diffusion (Stability.AI, 2022; Saharia et al., 2022; Rombach et al., 2022)), and transformers (*e.g.*, PixArt-$\alpha$ (Chen et al., 2023), FLUX (Black, 2024)) have demonstrated impressive capabilities in generating high-quality images from text prompts. These advancements have paved the way for the development of new generative image morphing techniques. Specifically, Wang & Golland (2023) leverages the local linearity of CLIP-based text embeddings to create smooth transitions between input images by interpolating the latent image features; Building upon this idea, IMPUS (Yang et al., 2023) introduces a multi-phase training framework that includes the optimization of text embeddings and training of Low-rank Adaptation (LoRA) modules to capture the semantics better. While this method yields more visually appealing results, it requires extensive training, typically around 30 minutes for each case. DiffMorpher (Zhang et al., 2024) proposes to directly interpolate the latent noise and leverage

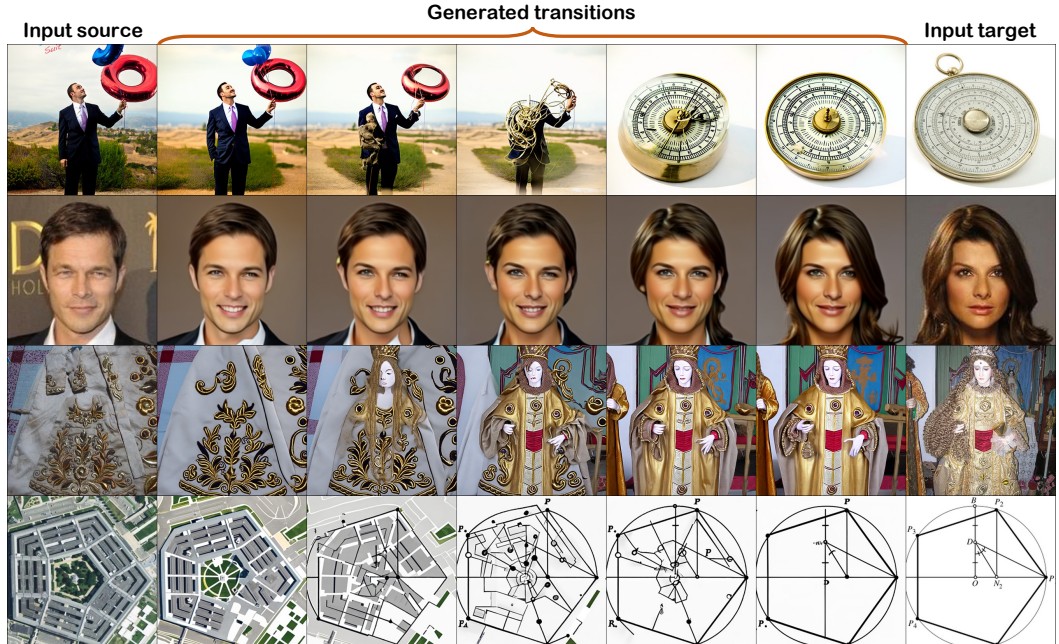

**Input source**      **Generated transitions**      **Input target**

Figure 1: **Examples of image morphing obtained via FreeMorph.** Given two input images, FreeMorph effectively generates smooth transitions between them within 30 seconds.

Adaptive Instance Normalization (AdaIN) to improve performance. However, they still encounter significant challenges in processing images with diverse semantics and intricate layouts, which limits their effectiveness in practical applications.

Given these issues, our objective is to accomplish image morphing without requiring further tuning. Nonetheless, this goal introduces two key challenges: **1) Inconsistent transition and identity loss.** While using a pre-trained diffusion model to convert input images into latent features and applying spherical interpolation might seem straightforward, this approach often results in inconsistent transitions. This is due to the non-linear characteristics of the multi-step denoising process. Additionally, this method inherits biases from the pre-trained model, which can lead to a loss of identity in the generated images. **2) Limitations in directional transitions.** The diffusion model does not inherently provide an effective "motion flow" to capture the gradual changes between images. As a result, without additional adjustments, achieving smooth and gradual transitions in a tuning-free manner remains a significant challenge.

In this paper, we present *FreeMorph*, a novel tuning-free method that is capable of generating directional and realistic transitions between two images instantly. Our method comprises two novel components: **1) Guidance-aware spherical interpolation:** We first enhance the pre-trained diffusion model by incorporating explicit guidance from the input images through modifications to the self-attention modules. This is achieved through spherical interpolation, which produces intermediate features used in two key ways. Firstly, we perform *spherical feature aggregation* to blend the key and value features from the self-attention modules, ensuring consistent transitions across the generated image sequences. Additionally, to address the issue of identity loss, we introduce an *prior-driven self-attention mechanism* which incorporates explicit guidance from the input images to preserve their unique identities. **2) Step-oriented motion flow:** To address the limitation in directional transitions, we introduce a novel step-oriented motion flow. This method blends two self-attention modules respectively derived based on each input image, enabling a controlled and directional transition that respects both input images. To further improve the quality of the generated image sequences, we have crafted an improved reverse diffusion and forward denoising process, seamlessly integrating these innovative components into the original DDIM framework. As shown in Fig. 1 and Fig. 4, our approach adeptly handles diverse input types, whether they share similar or distinct semantics/layouts, producing smooth and realistic transitions between images.

To thoroughly assess FreeMorph and benchmark it against current methods, we further develop a comprehensive evaluation system. This system includes **1)** four distinct sets of image pairs categorized

by their semantic and layout similarity, and **2)** specialized metrics designed to measure the smoothness, fidelity, and directness of image morphing methods. Our extensive evaluations demonstrate that FreeMorph substantially outperforms existing approaches. FreeMorph can produce high-fidelity image sequences with smooth and coherent transformations in under 30 seconds, which is $50\times$ faster than IMPUS (Yang et al., 2023) and $10\times$ faster than DiffMorpher (Zhang et al., 2024).

## 2 RELATED WORK

**Text-to-Image Generation.**   Recently, diffusion models (Rombach et al., 2022; Podell et al., 2023; Saharia et al., 2022; Ramesh et al., 2022) have emerged as the *de facto* method for text-to-image generation tasks. These models employ a series of denoising steps (DDIM, DDPM) (Ho et al., 2020; Song et al., 2021) to transform Gaussian noise into images, effectively capturing and interpreting the details conveyed by textual prompts. Moreover, trained on billions of text-image pairs(Schuhmann et al., 2022), these methods exhibit a remarkable ability to understand the distribution of real-world images, generating high-quality and diverse outputs while maintaining generalization. Our work harnesses the powerful capabilities of diffusion models, particularly their ability to generate smooth transition sequences between two specified images (Samuel et al., 2024; Khrulkov et al., 2023; He et al., 2024) to address the image morphing task.

**Image Morphing.**   Image morphing is a long-standing computer vision and graphics problem. Before the deep learning era, techniques such as mesh warping (Smythe, 1990; Wolberg, 1990; Fant, 1986) and field morphing (Beier & Neely, 1992; Lee et al., 1998) were the primary approaches employed in this domain. Recently, advancements in diffusion models have led to significant progress, as demonstrated by methods like DiffMorpher (Zhang et al., 2024), IMPUS (Yang et al., 2023), and Wang & Golland (2023). These approaches focus on optimizing text embeddings for two images and fine-tuning pre-trained text-to-image diffusion models to achieve smooth interpolation between them. However, they often necessitate extensive fine-tuning for each image pair and are limited to images with similar semantics and layouts. This can also hinder the generalizability of the pre-trained diffusion models due to the constraints imposed by LoRA modules in the U-Net architecture. In contrast, our method offers a tuning-free framework that makes no additional modifications to the original diffusion models, thereby preserving their inherent generalizability. Additionally, our approach significantly enhances training efficiency and is capable of handling images with varying layouts and semantics, addressing a challenging aspect for existing techniques.

**Tuning-Free Text-Guided Image Editing.**   Recent image translation methods have emerged that edit either generated or real-world images through text in a training-free manner, without altering the internal computation of the U-Net. For instance, SDEdit (Meng et al., 2022) proposes a straight-forward method to add $T$ time steps of Gaussian noise to the original images and denoise them using the guided text. Conversely, EDICT (Wallace et al., 2023) and FPI (Meiri et al., 2023) focus on inverting the reference image back to the latent space and subsequently applying the inverted latent condition guided by text. Additionally, methods like P2P (Hertz et al., 2023), PnP (Tumanyan et al., 2023), and MasaCtrl (Cao et al., 2023) modify the attention mechanism within diffusion models to enhance the alignment between guided text and the consistency of generated images with their original counterparts. Drawing inspiration from these existing techniques, our method aims to facilitate image morphing in a tuning-free manner. Notably, our approach also achieves comparable image editing performance by framing text-guided editing as a special case of morphing between real and generated images.

## 3 METHODOLOGY

Given two independent images $\mathcal{I}_{\text{left}}$, $\mathcal{I}_{\text{right}}$ as input, our objective is to generate a sequence of intermediate images $\mathcal{S} = \{\mathcal{I}_j\}_{j=1}^{J}$ that smoothly transform from one image to the other in a tuning-free manner. Note that we set $J = 5$ for experiments reported in this paper. As illustrated in Algorithm 1, our pipeline employs the pre-trained diffusion model as our foundation and integrates guidance from the input images into the multi-step denoising process. In the subsequent sections, we first introduce the preliminaries that underpin our method in Sec. 3.1. Next, we will describe the FreeMorph framework in detail. This illustration comprises three main components: 1) the

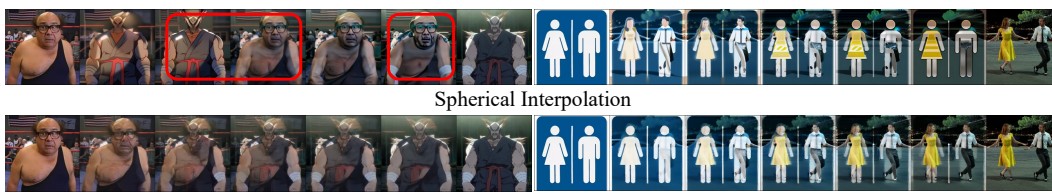
Spherical Interpolation

Replace Key, Value features

Figure 2: **Replacing the key and value feature in the attention mechanism.** We can observe that good key and value features would lead to smooth transitions and identity preservation.

guidance-aware spherical interpolation outlined in Sec 3.2, which includes our proposed spherical feature aggregation and prior-driven self-attention mechanism, 2) step-oriented motion flow that enables a controlled and directional image morphing (see Sec. 3.3), and 3) our improved reverse diffuison and forward denoising processes depicted in Sec. 3.4.

## 3.1 PRELIMINARIES

**Denoising Diffusion Implicit Model (DDIM).** DDIM (Song et al., 2021), trained on extensive text-image paired datasets, is designed to recover images from noisy inputs. It involves two main processes: 1) a series of reverse diffusion steps $\{q(\mathbf{z}_t)|t = 0, 1, ..., T\}$ that progressively add noise to the data, and 2) corresponding forward denoising steps $p(\mathbf{z}_t)|t = T, T-1, ..., 0$ that reconstruct clean data from the Gaussian noise. Here, $\mathbf{z}_t$ represents the latent features, and $T$ indicates the number of diffusion steps.

Once trained, DDIM provides a deterministic mapping between $\mathbf{z}_0$ and $\mathbf{z}_t$ through the reverse diffusion model $q(\mathbf{z}_t \mid \mathbf{z}_0) := \mathcal{N}\left(\mathbf{z}_t; \sqrt{\beta_t}\mathbf{z}_0, (1 - \beta_t)\mathbf{I}\right)$ and a parameterized noise estimator $\epsilon_\theta(\mathbf{x}_t, t)$. The relationship between these variables is given by:

$$\frac{\mathbf{z}_{t+1}}{\sqrt{\beta_{t+1}}} - \frac{\mathbf{z}_t}{\sqrt{\beta_t}} = \left(\sqrt{\frac{1 - \beta_{t+1}}{\beta_{t+1}}} - \sqrt{\frac{1 - \beta_t}{\beta_t}}\right)\epsilon_\theta^{(t)}(\mathbf{z}_t). \tag{1}$$

In practice, the noise estimator $\epsilon_\theta(\mathbf{x}_t, t)$ is typically implemented using a UNet (Ronneberger et al., 2015).

**Latent Diffusion Model (LDM).** Building upon DDIM, the Latent Diffusion Model (LDM) (Rombach et al., 2022) is a refined variant of diffusion models that effectively balances image quality with denoising efficiency. Specifically, LDM utilizes a pre-trained variational auto-encoder (VAE) (Kingma & Welling, 2013) to map images into a latent space and then trains the diffusion model within this space. Furthermore, LDM enhances the UNet architecture by incorporating self-attention modules, cross-attention layers, and residual blocks to integrate text prompts as conditional inputs during image generation. The attention mechanism in LDM's UNet can be formulated as:

$$\texttt{ATT}(Q, K, V) = \text{softmax}\left(\frac{Q \cdot K^T}{\sqrt{d_k}}\right) \cdot V \tag{2}$$

where $Q$ denotes the query features from spatial data, and $K$ and $V$ are key and value features derived from either spatial data (for self-attention) or text embeddings (for cross-attention). The noise estimator in LDM is then extended to $\epsilon_\theta(\mathbf{x}_t, t, y)$, where $y$ denotes the text embedding.

Our approach builds upon the Stable Diffusion model (Stability.AI, 2022), a pre-trained LDM developed by StabilityAI, and utilizes a vision-language model (VLM), LLaVA (Liu et al., 2024), for generating captions for the input images.

## 3.2 GUIDANCE-AWARE SPHERICAL INTERPOLATION

Existing image morphing methods (Nri, 2022; Zhang et al., 2024; Yang et al., 2023) typically involve training Low-rank Adaptation (LoRA) modules for each input image to enhance semantic comprehension and achieve smooth transitions. However, this approach is often inefficient and time-consuming and struggles with images that differ in semantics or layout. In this paper, we propose a tuning-free image morphing approach built on the pre-trained Stable Diffusion model.

**Algorithm 1** FreeMorph

**Input:** $\mathcal{I}_{\text{left}}, \mathcal{I}_{\text{right}}$

1: Caption the input images via pre-trained LLaVA $\rightarrow$ Text$_{\text{left}}$, Text$_{\text{right}}$.
2: Obtain image features $\mathbf{z}_{0-\text{left}}, \mathbf{z}_{0-\text{right}}$, and text embedding $y_{\text{left}}, y_{\text{right}}$ via VAE and text encoder of pre-trained Stable Diffusion.
3: Applying spherical interpolation to obtain $\mathbf{z}_{0-j}$ where $j \in [1, J]$ as initialization.
4: Reverse diffusion steps:
**for** $t = 1$ to $T$ **do**
  **if** $t < \lambda_1 \cdot T$ **then**
    Apply the original attention mechanism.
  **else if** $t < \lambda_2 \cdot T$ **then**
    Apply the prior-driven self-attention mechanism as in Eq. 5.
  **else**
    Apply the step-oriented motion flow as in Eq. 6.
  **end if**
**end for**
5: High-frequency Gaussian noise injection.
6: Forward denoising steps:
**for** $t = 1$ to $T$ **do**
  **if** $t < \lambda_3 \cdot T$ **then**
    Apply the step-oriented motion flow as in Eq. 6.
  **else if** $t < \lambda_4 \cdot T$ **then**
    Apply the spherical feature aggregation as in Eq. 4.
  **else**
    Apply the original attention mechanism.
  **end if**
**end for**
7: Add text-conditioned features.

**Output:** $J$ intermediate images gradually change from $\mathcal{I}_{\text{left}}$ to $\mathcal{I}_{\text{right}}$.

By leveraging the capabilities of DDIM for image inversion and interpolation, one might consider converting the input images ($\mathcal{I}_{\text{left}}, \mathcal{I}_{\text{right}}$) into latent features ($\mathbf{z}_{0-\text{left}}, \mathbf{z}_{0-\text{right}}$) and applying spherical interpolation may seem like a simple straightforward solution:

$$\mathbf{z}_{0-j} = \frac{\sin((1-j)\cdot\phi)}{\sin\phi} \cdot \mathbf{z}_{0-\text{left}} + \frac{\sin(j\cdot\phi)}{\sin\phi} \cdot \mathbf{z}_{0-\text{right}}, \tag{3}$$

where $j \in [1, J]$ is the index of intermediate images, and $\phi = \arccos(\frac{\mathbf{z}_{0-\text{left}}^T \cdot \mathbf{z}_{0-\text{right}}}{||\mathbf{z}_{0-\text{left}}||\cdot||\mathbf{z}_{0-\text{right}}||})$. Recall that we set $J = 5$ in our paper. However, directly inverting these interpolated latent features $\mathbf{z}_{0-j}$ to generate images often results in inconsistent transitions and identity loss (see Fig. 2). This issue arises because (1) the multi-step denoising process is highly non-linear, leading to discontinuous image sequences, and (2) there is no explicit guidance to control the denoising, causing the model to inherit biases from the pre-trained diffusion model.

**Spherical Feature Aggregation.** Drawing on insights from previous image editing techniques (Cao et al., 2023; Hertz et al., 2023; Parmar et al., 2023; Shi et al., 2024; Tumanyan et al., 2023), we observed that using the features $\mathbf{z}_{0-j}$ as initialization and replacing the key and value features ($K$ and $V$) in the attention mechanism (as described in Eq. 2) with features from the right image $\mathcal{I}_{\text{right}}$ can largely enhance the smoothness and identity preservation of the image transitions, although some imperfections may remain (see Fig. 2). Motivated by this finding, and recognizing that the query features ($Q$) largely reflect the overall image layout, we propose first blending features from both the left and right images ($\mathcal{I}_{\text{left}}, \mathcal{I}_{\text{right}}$) to provide explicit guidance for the multi-step denoising process. Specifically, in the denoising step $t$, we first feed the latent of the input images $\mathbf{z}_{t-\text{left}}$ and $\mathbf{z}_{t-\text{right}}$ to the pre-trained UNet $\epsilon_\theta$ to obtain the key and value features. Following that, We then substitute the original $K$ and $V$ with those derived from the input images and compute their average to modify the attention mechanism:

$$\text{ATT}(Q_{t-j}, K_{t-j}, V_{t-j}) := \frac{1}{2} \cdot (\text{ATT}(Q_{t-j}, K_{t-\text{left}}, V_{t-\text{left}}) + \text{ATT}(Q_{t-j}, K_{t-\text{right}}, V_{t-\text{right}})) \tag{4}$$

Latent noise inverted from

Noise distortion rate    0.00%    30.56%    65.97%    82.64%    97.22%    99.31%    100.00%

white jeep, black car, parked on street, near building, stop sign

Figure 3: **Effectiveness of the latent noise on the generated images.** The pre-trained diffusion model is robust to the noise distortion within the latent space.

where $Q_{t-j}, K_{t-j}, V_{t-j}$ are obtained by inputting $\mathbf{z}_{t-j}$ to the pre-trained UNet $\epsilon_\theta$. Note that $\mathbf{z}_{t-j}$, $\mathbf{z}_{t-\text{left}}$ and $\mathbf{z}_{t-\text{right}}$ are derived based on Eq. 2.

**Prior-driven Self-attention Mechanism.** While our feature blending technique significantly improves identity preservation in image morphing, we found that using this approach uniformly in both reverse diffusion and forward denoising stages can result in transitions where the image sequences change minimally and fail to accurately represent the input images (see Fig. 6). This outcome is anticipated because the latent noise will largely influence the forward denoising process, as shown in Fig. 3. Consequently, applying our feature blending, depicted in Eq. 4, introduces ambiguity as the consistent and strong constraints from the input images cause each latent noise $i$ to appear similar, thereby limiting the effectiveness of the transitions. To tackle this issue, we further propose a prior-driven self-attention mechanism that prioritizes the latent features from spherical interpolation to ensure smooth transitions within the latent noise, while emphasizing the input images to maintain identity preservation afterward. Specifically, during the forward denoising stage, we use the approach described in Eq. 4, while for the reverse diffusion steps, we employ a different attention mechanism as follows by modifying the self-attention modules:

$$\text{ATT}(Q_{t-j}, K_{t-j}, V_{t-j}) := \frac{1}{J} \sum_{k=1}^{J} \text{ATT}(Q_{t-j}, K_{t-k}, V_{t-k}) \tag{5}$$

Refer to Sec. 4.3 for detailed ablation studies on this design.

## 3.3 Step-oriented Motion Flow

After obtaining image sequences that are both coherent and accurately reflect the input identities, the next challenge is to achieve a smooth and gradual transition from the left image $\mathcal{I}_{\text{left}}$ to the right image $\mathcal{I}_{\text{right}}$. This problem stems from the lack of a "motion flow" that captures the changes from $\mathcal{I}_{\text{left}}$ to $\mathcal{I}_{\text{right}}$. To this end, we propose a step-oriented motion flow that gradually changes the influence between the input images ($\mathcal{I}_{\text{left}}$ and $\mathcal{I}_{\text{right}}$):

$$\text{ATT}(Q_{t-j}, K_{t-j}, V_{t-j}) := (1-\alpha_j) \cdot \text{ATT}(Q_{t-j}, K_{t-left}, V_{t-left}) + \alpha_j \cdot \text{ATT}(Q_{t-j}, K_{t-right}, V_{t-right}), \tag{6}$$

where $\alpha_j = j/(J+2-1)$, with $J+2$ representing the total number of images, which includes the $J$ generated images and the 2 input images.

## 3.4 Reverse diffusion and forward denoising process

**High-frequency Gaussian Noise Injection.** As discussed earlier, FreeMorph incorporates features from both the left and right images during the reverse diffusion and forward denoising stages. Nevertheless, we have observed that this can occasionally impose overly stringent constraints on the generation process. To mitigate this issue and allow for greater flexibility, we propose introducing Gaussian noise into the latent vector $\mathbf{z}$ in the high-frequency domain after the reverse diffusion steps:

$$\mathbf{z} := \begin{cases} \text{IFFT}(\text{FFT}(\mathbf{z})), & \text{if } \mathbf{m} = 1 \\ \text{IFFT}(\text{FFT}(\mathbf{g})), & \text{if } \mathbf{m} = 0 \end{cases} \tag{7}$$

Here, $\text{IFFT}(\cdot)$ and $\text{FFT}(\cdot)$ denote the inverse fast Fourier transform and fast Fourier transform, respectively. $\mathbf{g} \sim \mathcal{N}(0,1)$ represents a randomly sampled noise vector, and $\mathbf{m}$ is a binary high-pass filter mask of the same size as $\mathbf{z}$.

To enhance the efficacy of our image morphing process, we have found that consistently applying either guidance-aware spherical interpolation or step-oriented motion flow across all denoising steps

Table 1: **Quantitative comparison with existing image morphing techniques.**

| Method | MorphBench | | | Morph4Data | | | Overall | | |
|--------|------------|--|--|------------|--|--|---------|--|--|
| | LPIPS$_{sum}$ ↓ | FID$_{mean}$ ↓ | PPL$_{sum}$ ↓ | LPIPS$_{sum}$ ↓ | FID$_{mean}$ ↓ | PPL$_{sum}$ ↓ | LPIPS$_{sum}$ ↓ | FID$_{mean}$ ↓ | PPL$_{sum}$ ↓ |
| IMPUS Yang et al. (2023) | 130.52 | 152.43 | 3263.03 | 134.88 | 210.66 | 3199.90 | 265.40 | 174.76 | 6462.93 |
| DiffMorpher Zhang et al. (2024) | 90.57 | 157.18 | 2264.20 | 98.56 | 292.54 | 2394.05 | 189.13 | 209.10 | 4658.25 |
| Spherical Interpolation | 119.77 | 169.17 | 2994.35 | 103.74 | 245.22 | 2593.58 | 223.52 | 198.34 | 5587.93 |
| Ours | **84.91** | **141.32** | **2122.80** | **80.30** | **201.09** | **2007.52** | **162.99** | **152.88** | **4192.82** |

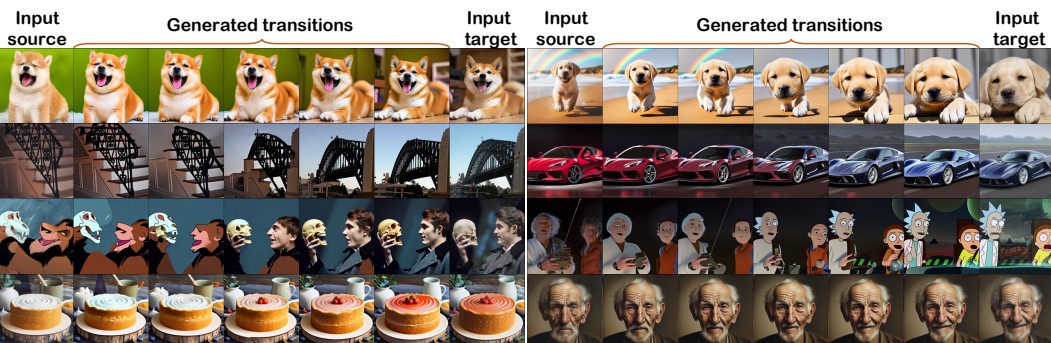

Figure 4: **More results produced by FreeMorph.** Our method can achieve smooth and high-fidelity image transitions for input images with either similar or different semantics and layouts.

yields suboptimal results (see Sec. 4.3). To address this, we have developed a refined approach for both reverse diffusion and forward denoising processes. We provide an overview algorithm of our proposed FreeMorph in Algorithm. 1. Specifically:

- **Reverse diffusion:** We use the standard self-attention mechanism for the first $\lambda_1 \cdot T$ steps. From $\lambda_1 \cdot T$ to $\lambda_2 \cdot T$, we apply the feature blending technique from Eq. 5. For the remaining steps, we implement the step-oriented motion flow.

- **Forward denoising:** We begin with the step-oriented motion flow for the first $\lambda_3 \cdot T$ steps, followed by the feature blending method from Eq. 4 for steps between $\lambda_3 \cdot T$ and $\lambda_4 \cdot T$. The process ends with the original self-attention mechanism for the final steps to produce images with higher fidelity.

Here, $\lambda_1$, $\lambda_2$, $\lambda_3$, and $\lambda_4$ are hyper-parameters and $T = 50$ is the total number of steps.

## 4 EXPERIMENTS

We now evaluate the performance of FreeMorph across various scenarios, comparing it with state-of-the-art image morphing techniques and conducting ablation studies to highlight the effectiveness of our proposed methods.

**Implementation Details.** We utilize version 2.1 of the publicly available Stable Diffusion model. Both the reverse diffusion and forward denoising processes use a DDIM schedule with $T = 50$ steps. Following the setting Stable Diffusion, we operate on the image resolution of $768 \times 768$. We configure the classifier-free guidance (CFG) parameter to 7.5 to incorporate text-conditioned features. The hyperparameters are set as follows: $\lambda_1 = 0.3$, $\lambda_2 = 0.6$, $\lambda_3 = 0.2$, $\lambda_4 = 0.6$. Additional implementation details can be found in the Appendix.

**Evaluation Datasets.** DiffMorpher (Zhang et al., 2024) introduces MorphBench, which includes 24 animation pairs and 66 image pairs, predominantly featuring images with similar semantics or layouts. To complement this and mitigate potential biases, we present **Morph4Data**, a newly curated evaluation dataset comprising four categories: 1) *Class-A* consisting of 25 image pairs with similar layouts but differing semantics, sourced from Wang & Golland (2023); 2) *Class-B* containing image pairs with both similar layouts and semantics, including 11 pairs of faces from CelebA-HQ (Karras et al., 2018) and 10 pairs of various car types; 3) *Class-C* featuring 15 pairs of randomly sampled images from ImageNet-1K (Deng et al., 2009) with no semantic or layout similarity.; 4) *Class-D* comprising 15 pairs of dog and cat images randomly sampled from the internet.

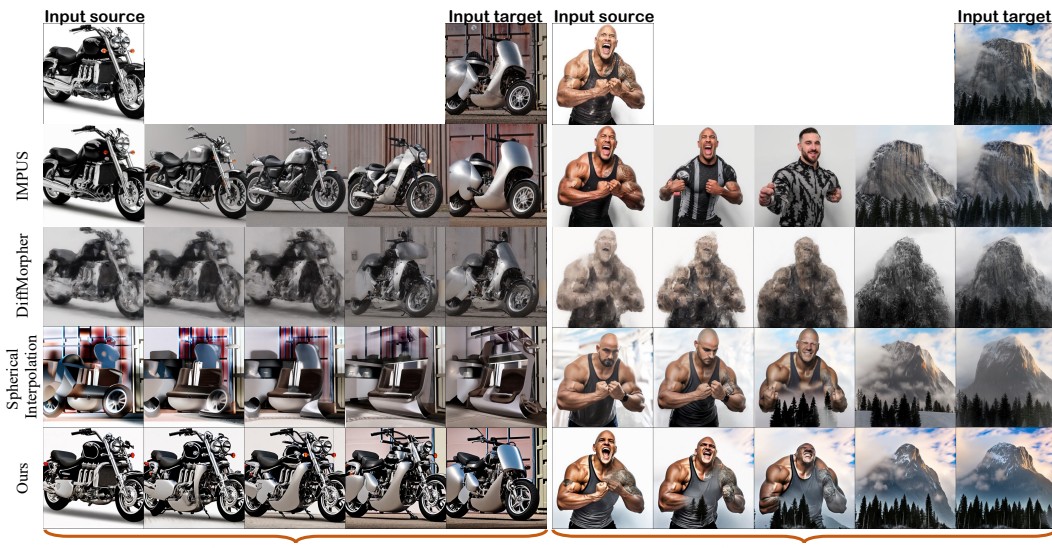

Figure 5: **Qualitative comparison with existing image morphing techniques.** Unlike other methods that struggle or fail to generate smooth and high-fidelity results without identity loss, our approach consistently achieves high-quality transitions, yielding superior results.

## 4.1 QUANTITATIVE EVALUATIONS

Following IMPUS (Yang et al., 2023) and DiffMorpher (Zhang et al., 2024), we conducted additional quantitative comparisons using the following metrics: 1) Frechet Inception Distance (FID) (Heusel et al., 2017), which assesses the similarity between the distributions of input and generated images; 2) Perceptual Path Length (PPL) (Karras et al., 2020), where we calculate the sum of PPL loss between adjacent images; and 3) Learned Perceptual Image Patch Similarity (LPIPS) (Zhang et al., 2018), which we also sum for adjacent images to evaluate the smoothness and coherence of the generated transitions. The results, detailed in Table 1, demonstrate the superior performance of our image morphing network across both datasets, showing enhanced fidelity, smoothness, and directness in the generated transitions.

## 4.2 QUALITATIVE EVALUATIONS

**More Qualitative Results.** In Fig. 1 and Fig. 4, we present a wide range of results produced by FreeMorph, consistently showcasing its ability to generate high-quality and smooth transitions. Our method excels across diverse scenarios, accommodating images with varying semantics and layouts, as well as those with similar characteristics. Additionally, FreeMorph effectively handles subtle variations, such as cakes with different colors and individuals with different expressions.

**Qualitative Comparisons.** We provide qualitative comparisons with existing image morphing methods in Fig. 5. The key observations are as follows: **1)** When handling images with varying semantics and layouts, IMPUS (Yang et al., 2023) exhibits identity loss and produces unsmooth transitions; **2)** Although Diffmorpher (Zhang et al., 2024) achieves smoother transitions compared to IMPUS, its results often suffer from blurriness and lower overall quality; **3)** We also evaluate a baseline approach, which involves simply applying the spherical interpolation and DDIM process to obtain the results. Through the visualizations, we notice that this baseline approach experiences (i) challenges in accurately interpreting the input images due to the absence of explicit guidance, and (ii) suboptimal results in terms of image quality. In contrast, our method consistently delivers superior performance, characterized by smoother transitions and higher image quality. Additional comparisons are available in the Appendix.

## 4.3 FURTHER ANALYSIS

**Analysis of Guidance-aware Spherical Interpolation.** In Fig. 6, we present ablation studies to evaluate the effects of the proposed spherical feature aggregation (described in Eq. 4) and prior-driven

Table 2: **Quantitative comparison for ablation studies.**

| Method | MorphBench | | | Morph4Data | | | Overall | | |
|---|---|---|---|---|---|---|---|---|---|
| | LPIPS$_{sum}$ ↓ | FID$_{mean}$ ↓ | PPL$_{sum}$ ↓ | LPIPS$_{sum}$ ↓ | FID$_{mean}$ ↓ | PPL$_{sum}$ ↓ | LPIPS$_{sum}$ ↓ | FID$_{mean}$ ↓ | PPL$_{sum}$ ↓ |
| w/ only Eq. 5 | 157.01 | 320.05 | 3425.19 | 141.12 | 411.80 | 3028.05 | 298.13 | 355.24 | 6453.24 |
| w/ only Eq. 4 | 99.69 | 155.51 | 2491.10 | 90.80 | 217.26 | 2270.05 | 190.49 | 179.20 | 4761.15 |
| w/ only Eq. 5 and Eq. 4 | 211.52 | 243.08 | 5288.10 | 139.55 | 290.11 | 3488.87 | 351.08 | 261.12 | 8776.96 |
| w/o noise injection | 99.49 | 154.53 | 2487.16 | 89.12 | 211.23 | 2228.03 | 188.61 | 176.28 | 4715.19 |
| w/o Eq. 4 | 87.41 | 155.46 | 2185.30 | 81.10 | 218.95 | 2027.58 | 168.52 | 179.82 | 4212.88 |
| w/o Eq. 5 | 120.01 | 148.54 | 3000.35 | 101.28 | 215.43 | 2572.06 | 221.30 | 174.19 | 5572.41 |
| w/o step-oriented motion flow | 118.50 | 154.71 | 2962.48 | 93.39 | 214.93 | 2334.68 | 211.89 | 177.80 | 5297.17 |
| Ours (Var-A) | 153.40 | 184.54 | 3835.08 | 115.91 | 243.20 | 2897.63 | 269.31 | 207.04 | 6732.70 |
| Ours (Var-B) | 93.54 | 158.44 | 2338.62 | 85.76 | 245.36 | 2144.08 | 179.31 | 191.78 | 4482.70 |
| Ours | **84.91** | **141.32** | **2122.80** | **80.30** | **201.09** | **2007.52** | **162.99** | **152.88** | **4192.82** |


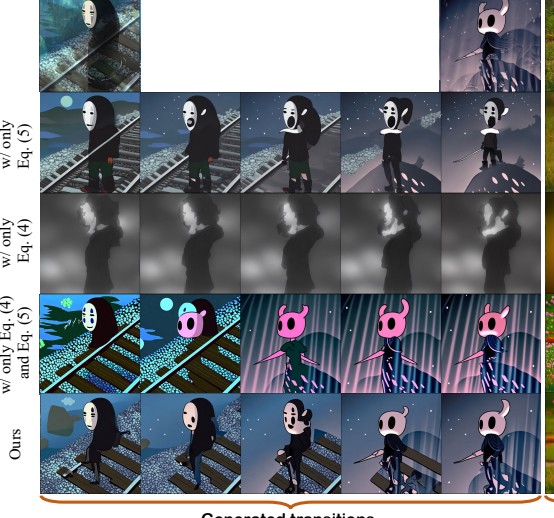
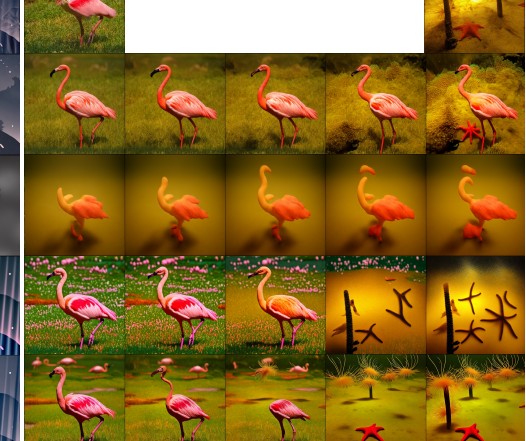

Figure 6: **Analysis of guidance-aware spherical interpolation.**

self-attention mechanism (as in Eq. 5) within our method. The results indicate that using either component alone yields suboptimal outcomes. Specifically, (i) spherical feature aggregation is crucial for achieving directional transitions where the characteristics of $\mathcal{I}_{left}$ should gradually diminish; (ii) the prior-driven self-attention mechanism is vital for preserving identity in the generated images. The combination of both components allows FreeMorph to produce smooth transitions while effectively maintaining identity. Furthermore, by comparing the last two rows in Fig.6, we demonstrate the importance of our step-oriented motion flow and the crafted reverse and forward processes.

**Analysis of Reverse and Forward Process.** In Fig. 7, we evaluate our method against two variants: (i) "Ours (Var-A)", which omits the original attention mechanism, and (ii) "Ours (Var-B)", which swaps the employed steps of the guidance-aware spherical interpolation and the step-oriented motion flow in both the reverse and forward processes. Comparing these variants with our final design reveals that (i) the original attention mechanism is crucial for achieving high-fidelity results, and (ii) the specific configuration of the reverse and forward processes in our final design yields optimal performance.

**Analysis of Step-oriented Motion Flow.** In Fig. 8, we first disable our proposed step-oriented motion flow to assess its impact. We observe that without this component, the model tends to produce abrupt changes rather than smooth transitions in the generated images. Additionally, the ending point, *i.e.*, the last generated image, exhibits high-contrast colors that differ from the target image $\mathcal{I}_{right}$. In comparison, the step-oriented motion flow enables our method to achieve smoother transitions and results in an ending point that is more closely aligned with the target image.

**Analysis of High-frequency Noise Injection.** We then disable high-frequency noise injection and present the ablation studies in Fig. 8. The results indicate that incorporating our proposed high-frequency noise injection enhances the model's flexibility and contributes to smoother transitions.

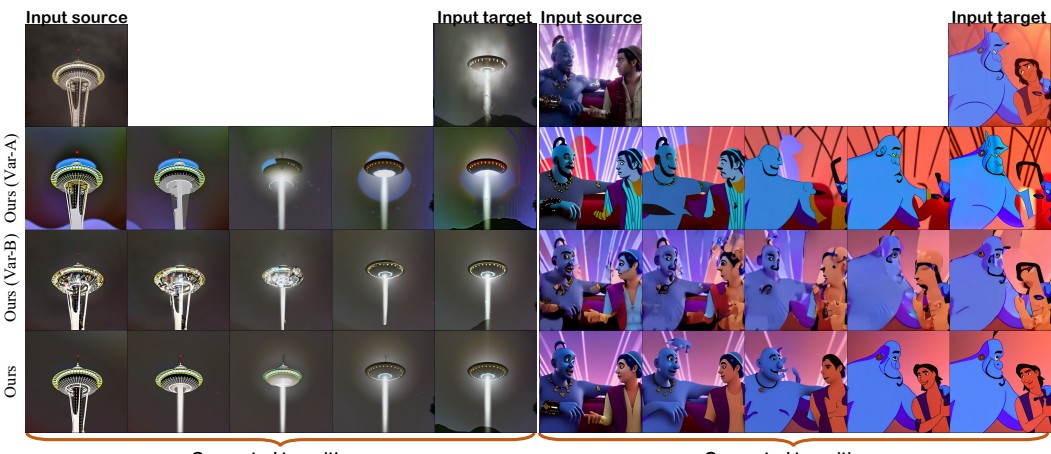

Figure 7: **Analysis of reverse diffusion and forward denoising process.**

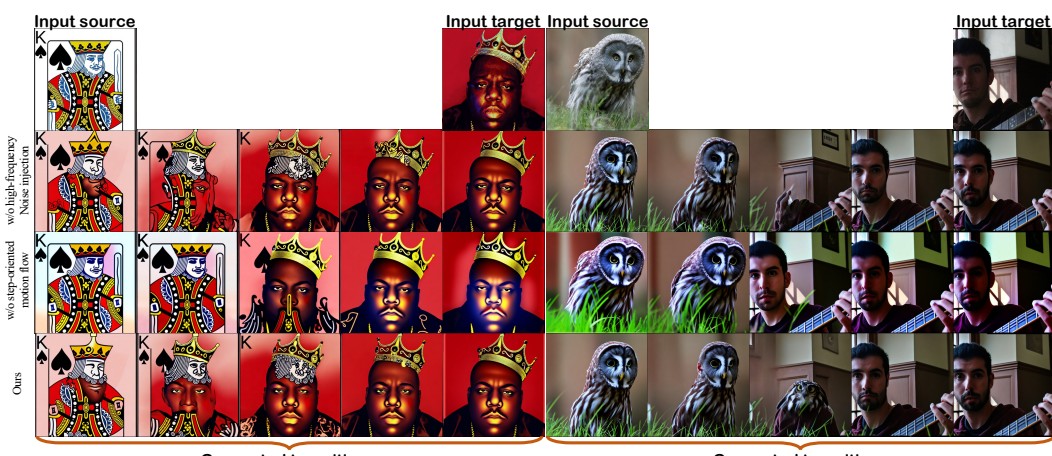

Figure 8: **Analysis of high-frequency noise injection and step-oriented motion flow.**

**Limitations and Failure Cases.** While establishing a new state-of-the-art, we recognize that our model has certain limitations. We illustrate several failure cases in Fig.9. Specifically: **1)** Although our model can achieve reasonable results when processing images that have no similarity in semantics and layouts, the generated transitions may not be smooth, potentially leading to abrupt changes during the transition; and **2)** Our method inherits biases from Stable Diffusion(Stability.AI, 2022), leading to difficulties in accurately transitioning images that model the limbs of human subjects.

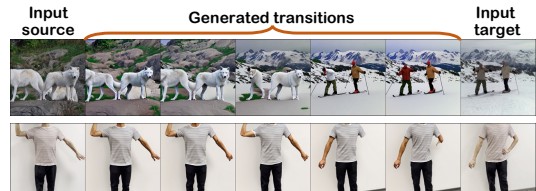

Figure 9: **Failure cases.**

## 5  CONCLUSION

We have introduced FreeMorph, a novel tuning-free pipeline that is capable of generating smooth and high-quality transitions between two input images within 30 seconds. Specifically, we propose to incorporate explicit guidance from the input images through modifications to the self-attention modules. This is achieved by two novel components, *i.e.*, spherical feature aggregation and prior-driven self-attention mechanism. Additionally, we introduce a step-oriented motion flow that ensures directional transitions consistent with both input images. We also carefully designed an improved reverse diffusion and forward denoising process to integrate our proposed modules into the original DDIM framework. Extensive experiments reveal that FreeMorph delivers high-fidelity results across various scenarios, significantly outperforming existing image morphing techniques.

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
