# OpenReview forum: "FreeMorph: Tuning-Free Generalized Image Morphing with Diffusion Model"
_ICLR.cc/2025/Conference — Submitted to ICLR 2025_

### Official Review · Reviewer_qmRm · 2024-10-30

**Soundness:** 3
**Presentation:** 3
**Contribution:** 3
**Rating:** 8
**Confidence:** 3

**Summary:**

This paper propose FreeMorph, a tuning-free pipeline that can generate transitions between two images within 30 seconds. It proposes two components, guidance-aware spherical interpolation, and Step-oriented motion flow, which yields smooth transitions between source and target images, even at examples of cross semantics. Both visual and quantitative evaluations are provided.

**Strengths:**

I think this paper is above the bar, with good results and reasonable techinques. Experiments are well designed, limitations are discussed.

**Weaknesses:**

Some motivation parts are not that clear to me, see below

**Questions:**

This paper targets at two issues related to the image morphing, identity loss and directional trasition. However, I do not see the importance of these two issues. Why they are important and worthing efforts for this task.

For example, when not preserving the identity, what results would be like. Similarly, what is the issue when miss preserve directional transition. I understand that the smooth transition is important, is it the same as the meaning of directional transition?

 These are important questions to me when reading the introduction. I can see cool examples in the intro, but cannot figure out clearly the core issues that this paper handles. I try to solve this by see examples of compared other methods, and some of them are still OK during the transition.

What is the successful rate, are the results cherry-picked

---

> ### Author Response · Authors · 2024-11-28
>
> # Response to reviewer `#qmRm`
>
>
> > This paper targets at two issues related to the image morphing, identity loss and directional trasition. However, I do not see the importance of these two issues. Why they are important and worthing efforts for this task.
>
> For instance, (1) For the second case in Fig. 5, IMPUS exhibits identity loss, where the third generated image deviates from the original identity; (2) In Fig. 16, IMPUS shows a lack of similarity between the second and third generated images, leading to unsmooth transitions.
>
> > I understand that the smooth transition is important, is it the same as the meaning of directional transition?
>
> We believe that these two terms are different. A smooth transition implies a gradual shift from the left image to the right image, maintaining continuity throughout. On the other hand, a directional transition signifies a progression where there is no loss of identity or regression to a previous state.
>
> > What is the successful rate, are the results cherry-picked
>
> Our successful rate is very high and the results are not cherry-picked. To reinforce this, we conducted user studies where 30 volunteers, comprising animators, AI researchers, and gaming enthusiasts aged between 20 and 35, participated. The outcomes of the user studies can be found below, showcasing the effectiveness of our proposed method through a subjective evaluation.
>
> |IMPUS|DiffMorpher|Slerp|Ours|
>
> |Preference|17.16%|14.89%|7.82%|**60.13%**

---

### Official Review · Reviewer_NcSG · 2024-10-30

**Soundness:** 3
**Presentation:** 1
**Contribution:** 2
**Rating:** 5
**Confidence:** 3

**Summary:**

This paper presents a novel tuning-free image morphing approach, which utilizes the pre-trained stable diffusion and manipulates its self-attention to achieve high-fidelity morphing. Specifically, it proposes a guidance-aware spherical interpolation and step-oriented motion flow, and respectively apply them in different steps of the reverse diffusion/forward denoising processes. Based on the qualitative and quantitative results, the proposed approach outperform the existing methods.

**Strengths:**

- The proposed tuning-free morphing approach is straightforward and effective. The modified interpolation of the self-attention is able to produce more natural transition between morphing sequences.
- A new dataset, Morph4Data, is presented and used in the evaluation. This dataset contains four categories, which helps in a detailed analysis on the image morphing methods.
- This paper presents and analyzes many baseline settings in the ablation study. This helps in understanding the specific role of each key design.

**Weaknesses:**

- This paper is apparently not well prepared for publication. For example:
  - There are many errors in spelling (“ur” in line 122, “owever” in line 133, etc.) and formulations (missing “{“ and “}” in line 179, etc.).
  - In equation (7), since m has the same size as z, what does it mean by “m=1”?
  - Both “DeepMorpher” and “deepmorpher” appear in the paper.
  - The image captions of Fig.2 and Fig.3 is too simple. Reader have to move to the main text to try to understand the figures.
- In the introduction, it mentions a comprehensive evaluation system, including a new dataset and specialized metrics. But in Sec 4.1, it says “following IMPUS and DiffMorpher, we ... using the following metrics”. Where are the proposed specialized metrics?
- This paper presents both the qualitative and quantitative evaluations. In the figures, the “spherical interpolation” looks good than the existing method DiffMorpher. But based on the quantitative evaluation in Table 1, “spherical interpolation” performs worse than DiffMorpher. Therefore, either the visual results are cherry-picked, or the evaluation metrics cannot adequately measure the results. Without a human perceptual study, it’s difficult to get a conclusion when comparing the two methods.
- The proposed algorithm contains a reverse diffusion process and a forward denoising process. In my understanding, the diffusion process of DDIM is to directly add noise without using the trained network. But in Algorithm 1, the modified attention mechanism is applied in the diffusion process. There should be more details to explain how to implement this diffusion process. In addition, why the forward denoising steps also have “for t=1 to T”?
- For the experiment “Ours(Var-A)” in line 469, what does it mean to “omit the original attention mechanism”? Completely remove the attention?

**Questions:**

My main concerns are the evaluation metrics and the technical details about the reverse diffusion process. I'd like to increase my rating if there are convincing explanations or evidences in the rebuttal.

---

> ### Author Response · Authors · 2024-11-28
>
> # Response to reviewer `#NcSG`
>
> > This paper is apparently not well prepared for publication.
>
> Thanks for the notification from the reviewer. We have rectified the identified typos and meticulously proofread the paper to ensure it is more refined and polished.
>
> > In equation (7), since m has the same size as z, what does it mean by “m=1”?
>
> It means that the value of each item in m is equal to 1.
>
> > The image captions of Fig.2 and Fig.3 is too simple. Reader have to move to the main text to try to understand the figures.
>
> Thanks for the suggestion. We have carefully added explanation for Fig.2 and Fig.3 in their captions.
>
> > In the introduction, it mentions a comprehensive evaluation system, including a new dataset and specialized metrics. But in Sec 4.1, it says “following IMPUS and DiffMorpher, we ... using the following metrics”. Where are the proposed specialized metrics?
>
> We have rectified the claim in the introduction. In this paper, we present an evaluation benchmark featuring a novel dataset comprising four unique sets of image pairs classified based on their semantic and layout similarities.
>
>
> > Without a human perceptual study, it’s difficult to get a conclusion when comparing the two methods.
>
> We have carried out user studies involving 30 volunteers, encompassing a diverse group of participants such as animators, AI researchers, and gaming enthusiasts, within the age range of 20 to 35. The results are provided below, showcasing the effectiveness of our proposed method through a subjective evaluation.
>
> |IMPUS|DiffMorpher|Slerp|Ours|
>
> |Preference|17.16%|14.89%|7.82%|**60.13%**
>
> > In my understanding, the diffusion process of DDIM is to directly add noise without using the trained network.
>
> Indeed, throughout the training process, the diffusion process will first generate random noise and then iteratively denoise the image to produce the clearer images.
>
> However, during the reverse diffusion steps, the network operates conversely by commencing with the image itself and utilizing the pre-trained UNet to progressively obtain the noise pattern corresponding to that image.
>
>
> > For the experiment “Ours(Var-A)” in line 469, what does it mean to “omit the original attention mechanism”? Completely remove the attention?
>
> In line 469, the term "Ours (Var-A)" denotes the approach that solely employs spherical interpolation (as in Eq. (3)) in both the reverse diffusion and forward denoising steps. To prevent any potential confusion, we will provide further clarification on this in the revised version.

---

> > ### Comment · Reviewer_NcSG · 2024-12-02
> >
> > Some of my concerns remain unaddressed after reading the authors' responses.
> >
> > First, there should be a convincing explanation about the inconsistency between the qualitative and quantitative comparison of "DiffMorpher" and "spherical interpolation".
> > >This paper presents both the qualitative and quantitative evaluations. In the figures, the “spherical interpolation” looks good than the existing method DiffMorpher. But based on the quantitative evaluation in Table 1, “spherical interpolation” performs worse than DiffMorpher. Therefore, either the visual results are cherry-picked, or the evaluation metrics cannot adequately measure the results. Without a human perceptual study, it’s difficult to get a conclusion when comparing the two methods.
> >
> > Second, there should be more information about the user study, i.e. how many results are shown to the volunteers?
> >
> > Third, I still don't understand the reverse diffusion steps. How does the pre-trained UNet obtain noise pattern based on the original natural image itself? The UNet is trained to predict noise from noisy images.

---

> ### Author Response · Authors · 2024-12-03
> **Response to Reviewer NcSG**
>
> Thank you gain for your constructive feedback.
>
> ***Q1: The explanation about the inconsistency between the qualitative and quantitative comparison of "DiffMorpher" and "spherical interpolation".***
>
> a) In Table 1, LPIPS, FID, and PPL are computed by extracting features of adjacent image pairs through the deep neural networks. While the results from "spherical interpolation" appear visually clearer than those from "DiffMorpher," "spherical interpolation" often generates semantically irrelevant content. For instance, as illustrated in Fig. 5, the images generated by "spherical interpolation" lose the semantic essence of the motorcycle. Additionally, there are significant changes between adjacent images, including both foreground and background elements, which result in larger feature variations between adjacent image pairs. In contrast, despite introducing noticeable image artifacts, "DiffMorpher" effectively preserves the semantic consistency between the source and target images. Furthermore, the transitions between adjacent images exhibit smoother background changes and maintain similar semantic content. These factors contribute to "DiffMorpher" outperforming "spherical interpolation" in Table 1.
>
> b) In the Supplementary Material, we provide additional generated results. These results show that for cases with similar semantics and layouts, "DiffMorpher" produces superior outputs compared to "spherical interpolation".  MorphBench, an evaluation dataset constructed by "DiffMorpher," is composed of images with similar semantics and layouts, which aligns well with the strengths of "DiffMorpher". Consequently, "DiffMorpher" achieves better performance than "spherical interpolation" on MorphBench. On the other hand, Morph4Data, our newly proposed evaluation dataset, includes cases with differing semantics and layouts. On Morph4Data, "spherical interpolation" achieves a better FID score than "DiffMorpher". This also demonstrate that our proposed evaluation datasets are more comprehensive.
>
> ***Q2: more information about the user study.***
>
> We provided a total of 50 examples to 30 volunteers for evaluation. Each example included the results of our method alongside the results generated by three other comparison methods.
>
> ***Q3: The reverse diffusion steps.***
> Diffusion inversion is a technique used in image edition and image inpainting. Its goal is to start from a given image $x_0$ and progressively "add noise" to revert it back to the initial noise state $x_T$.
>
> Here, we use the DDIM as the example for illustration. Instead of generating $x_{t-1}$ from $x_t$ via the DDIM inference formulation:
>
> $$
> x_{t-1} = \sqrt{\bar\alpha_{t-1}}(\frac{x_t-\sqrt{1-\bar\alpha_{t}}\epsilon_\theta(x_t)}{\sqrt{\bar\alpha_{t}}})+ \sqrt{1-\bar\alpha_{t-1}-\sigma_t^2}\epsilon_{\theta}(x_t) + \sigma_t\epsilon _t,
> $$
>
>
> we generate $x_t$ from $x_{t-1}$ using the reversed version of the above formula:
>
> $$
> x_t = \sqrt{\frac{\bar\alpha_t}{\bar\alpha_{t-1}}}x_{t-1}+\sqrt{\alpha_t} (\sqrt{\frac{1}{\alpha_t}-1 } - \sqrt{\frac{1}{\alpha_{t-1} } -1})\epsilon_{\theta}(x_{t-1}).
> $$
>
>
> Hence, by inputting $x_0$ into the pre-trained U-Net, we can obtain the predicted noise $\epsilon_{\theta}(x_{0})$. Using the above formula, we can then compute
> $x_1$. By iteratively applying this process, we can derive the initial noise state
> $x_T$ from a given image using the pre-trained U-Net. For more detailed information, please refer to [1, 2].
>
>
> [1] Edict: Exact diffusion inversion via coupled transformations. CVPR 2023
>
> [2] Prompt-to-prompt image editing with cross attention control. ICLR 2023

---

### Official Review · Reviewer_JPFT · 2024-11-03

**Soundness:** 2
**Presentation:** 2
**Contribution:** 2
**Rating:** 5
**Confidence:** 4

**Summary:**

This paper presents FreeMorph, a tuning-free method for image morphing that accommodates inputs with varying semantics and layouts. Several modules are proposed to enhance the quality of image interpolants, ensuring a smooth transition that remains consistent with the given images at both ends.

**Strengths:**

1. This method does not require training, making it more accessible to regular users.
1. The authors have conducted extensive experiments to identify the best combination of modules and hyperparameters.
1. Although this is a task that is difficult to evaluate, I can appreciate the authors’ efforts to provide both quantitative and qualitative results.

**Weaknesses:**

Related to the task
1. There are some unverified claims at the core of this task. For example, see lines 254-255.
1. What constitutes good image morphing is somewhat underdefined. The intended effect that this paper seeks to achieve feels too vague and abstract, making it challenging to identify a clear definition of effective image morphing.
1. Even with quantitative metrics like PPL, high scores do not necessarily indicate results that align with human preferences, which can be highly subjective.

Related to the proposed method
1. While the authors have introduced several modules, their use appears to have been determined empirically. The selection of hyperparameters and the final module composition could benefit from clearer justification, as they seem somewhat arbitrary at times.
1. Given that the method involves multiple components, the presentation could be more streamlined. It can be difficult to understand the specific problem each module aims to address. (Please see questions below for further clarification.)
1. The generated samples have pretty visible ghost artifacts. To me, that would not be considered as good image morphing.

**Questions:**

1. What's the intuition behind Equation 5. How does it relate to identity preservation?
1. How does Equation 6 connect to motion flow? What do you mean by "step-oriented"?

---

> ### Author Response · Authors · 2024-11-28
>
> # Response to reviewer `#JPFT`
>
> > There are some unverified claims at the core of this task. For example, see lines 254-255.
>
> In lines 254-255, we aim to emphasize that the multi-step denoising procedure is notably nonlinear, potentially leading to inconsistencies in the generated image morphing.
>
> To demonstrate the nonlinear nature of the multi-step denoising process, we would like to refer the reviewer to Eq. (9) in the DDIM paper[1]
>
> > What constitutes good image morphing is somewhat underdefined. The intended effect that this paper seeks to achieve feels too vague and abstract, making it challenging to identify a clear definition of effective image morphing.
>
> An effective image morphing outcome should exhibit gradual transitions from the initial (left) image to the final (right) image while preserving the original identities. For instance, (1) In the second example depicted in Fig. 5, IMPUS displays an identity loss where the third generated image deviates from the original identity; (2) Within Fig. 16, the lack of similarity between the second and third generated images results in abrupt transitions, detracting from smoothness.
>
> > Even with quantitative metrics like PPL, high scores do not necessarily indicate results that align with human preferences, which can be highly subjective.
>
> We concur with the reviewer's observation that existing quantitative metrics face challenges in accurately assessing result quality. In our research, we follow DiffMorpher and IMPUS to conduct PPL, LPIPS, and FID evaluations.
>
> To enhance our comparative analysis and include human preferences, we further conducted user studies. Specifically, we invited 30 volunteers encompassing animators, AI experts, and gaming enthusiasts aged between 20 and 35 to choose their most favorable results. The results are presented below, demonstrate the effectiveness of our proposed approach from a subjective standpoint.
>
> |IMPUS|DiffMorpher|Slerp|Ours|
>
> |Preference|17.16%|14.89%|7.82%|**60.13%**
>
>
> > While the authors have introduced several modules, their use appears to have been determined empirically. The selection of hyperparameters and the final module composition could benefit from clearer justification, as they seem somewhat arbitrary at times.
>
> In Fig. 2, our experiments involve substituting the key and value features within the attention mechanism. Through this analysis, we notice the pivotal role played by these features in facilitating seamless transitions and maintaining the inherent identity within the images. Consequently, the core enhancement in our design stems from the modifications made to the key and value features.
>
> The design outlined in Sections 3.2 and 3.3 is crafted to handle their respective issues, aiming to achieve optimal performance. Indeed, our pipeline's final iteration is a product of iterative experimentation. Despite its simplicity, our strategy proves effective and logical. Moreover, our approach surpasses existing techniques without necessitating extensive training.
>
> > What's the intuition behind Equation 5. How does it relate to identity preservation?
>
> In Figure 6, we observe that the absence of Eq. (5) can lead to challenges in generating smooth transitions in extreme scenarios. Instead, relying on the key and value features generated through spherical interpolation, as described in Eq. (3), leads to gradual changes (see Fig. 2). Therefore, the key lies in determining the optimal method for this replacement of key and value features.
>
> Through our experiments, we find that by averaging the results of spherical interpolations, we are able to achieve the best performance.
>
> > How does Equation 6 connect to motion flow? What do you mean by "step-oriented"?
>
> In our "step-oriented motion flow" process, we compute the gradual transition from the initial (left) image to the final (right) image. Initially, we opted to term it "motion flow" to signify this progression. However, upon reflection, we recognize that "motion flow" typically refers to changes in motion within 2D or 3D spaces, potentially causing confusion. As a result, we will adjust this terminology in the updated version.
>
> The alterations in attention features evolve progressively based on the step j. This is precisely why we coined it as "step-oriented."
>
> [1] Denoising Diffusion Implicit Models. ICLR 2021.

---

### Official Review · Reviewer_XAhp · 2024-11-03

**Soundness:** 3
**Presentation:** 2
**Contribution:** 2
**Rating:** 5
**Confidence:** 3

**Summary:**

The paper proposes a method for image interpolation with off-the-shelf T2I diffusion models. By modifying the attention modules using two proposed methods, the continuousness of the interpolations is improved, without having to fine-tune the diffusion model. The authors also introduce auxiliary tricks to further improve interpolation performance. The contribution of the various proposed components to improved interpolation performance is validated using an extensive ablation study, and the overall performance is compared qualitatively and quantitatively with some alternative approaches.

**Strengths:**

The quantiative and qualitative experiments, especially the ablation study seem well-executed and thorough, with the exception of one important method missing (see weaknesses). I really appreciate the authors thoroughly evaluating across different levels of challenging interpolations and proving a large number of qualitative comparisons in the appendix.

The proposed method seems reasonable in construction and effective in practice.

In the comparisons performed by the authors, their proposed method seems to provide a clear improvement over the methods they compare against.

Besides small mistakes (see questions), the paper is generally well-structured, reasonably well-written and easy to follow.

**Weaknesses:**

Attention Interpolation for Text-to-Image Diffusion (He et al., 2024) is cited in the related work section but never compared to, despite the methods seeming quite similar. Given that the authors were aware of this work at the time of submission, I would expect the authors to discuss how their and this method relate and incorporate it into quantitative and qualitative comparisons.

Similarly, Smooth Diffusion (Guo et al., CVPR 2024) is quite closely related and could also be compared to, although it requires additional tuning and thus seems a lot less relevant as a baseline. Demonstrating that the proposed method performs similarly to one requiring extra fine-tuning would strengthen the paper substantially.

**Questions:**

Eq. 3 looks wrong to me. Defining it with j as an integer in the range [1, 5] does not result in a slerp, as far as I know.

3.4: I think using a DCT instead of an FFT could help improve quality by getting rid of the correlation between the pixels on the left and right (and top and bottom respectively) edge of the image.

The number of decimal digits in tables is excessive. Please reduce it to a reasonable number (more than 3 or 4 digits should not be necessary in most cases) to aid readability.

There are a bunch of small mistakes. I'd recommend going over the paper and fixing them for a potential camera-ready version, as the current state looks a bit sloppy. Some examples: l. 122: "ur" -> "Our", l. 130: "Recently, Recently,"; l. 132: "wang2023interpolating"; l. 194: wrong citation for LDM, l. 211: I think DiffMorph should be attributed to different people.

---

> ### Author Response · Authors · 2024-11-28
>
> # Response to reviewer `#XAhp`
>
> > Attention Interpolation for Text-to-Image Diffusion (He et al., 2024) is cited in the related work section but never compared to, despite the methods seeming quite similar.
>
> > Similarly, Smooth Diffusion (Guo et al., CVPR 2024) is quite closely related and could also be compared to, although it requires additional tuning and thus seems a lot less relevant as a baseline.
>
> Thank you for the reviewer's suggestions. We have conducted experiments to compare our approach with the two methods mentioned and find that these method can only handle images with similar layout and semantics. Instead, our method can efficiently handle this case. Additionally, it's important to note that “Attention Interpolation for Text-to-Image Diffusion” relies on the IP-Adapter for image morphing, which compromises training efficiency. Furthermore, “Smooth Diffusion” necessitates tuning, making it slower and less efficient than our method. We will add more discussion and comparisons with these methods in the revised edition
>
> > Eq. 3 looks wrong to me. Defining it with j as an integer in the range [1, 5] does not result in a slerp, as far as I know.
>
> In our experiments, we configured the pipeline to generate five transition images. That's the reason why we compute five sampling points (with j as an integer in the range [1, 5]) in Eq. (3), each corresponding to a transition image. Moreover, our approach allows for setting j to any value. As the number j increases, Eq. (3) will increasingly resemble a more smooth slerp operation.
>
> > I think using a DCT instead of an FFT could help improve quality by getting rid of the correlation between the pixels on the left and right (and top and bottom respectively) edge of the image.
>
> Thank you for the suggestion. We have carried out experiments to contrast the DCT with the FFT and observe that DCT perform on par with FFT. This situation might because our Gausian noise inject doesn't help improve a lot for the result.
>
> > The number of decimal digits in tables is excessive. Please reduce it to a reasonable number (more than 3 or 4 digits should not be necessary in most cases) to aid readability.
>
> Thanks for the suggestion from the reviewer. We have made refinements to the decimal digit precision in Tab. 1 and Tab. 2.
>
>
> > There are a bunch of small mistakes. I'd recommend going over the paper and fixing them for a potential camera-ready version, as the current state looks a bit sloppy. Some examples: l. 122: "ur" -> "Our", l. 130: "Recently, Recently,"; l. 132: "wang2023interpolating"; l. 194: wrong citation for LDM, l. 211: I think DiffMorph should be attributed to different people.
>
> Thanks for the suggestion from the reviewer. We have diligently proofread the paper and rectified those minor errors in the revised edition.

---

> > ### Comment · Reviewer_XAhp · 2024-12-01
> >
> > Thank you for the response.
> >
> > Regarding the main concerns I mentioned, you mention that you performed additional experiments, but I cannot locate them either in the revised main PDF or in the supplementary material. Could you please point me to where you show the quantitative/qualitative results from these experiments?

---

> > > ### Author Response · Authors · 2024-12-03
> > >
> > > We sincerely thank the reviewer for the response and suggestions.
> > >
> > > Regrettably, due to the rush at the time, we inadvertently submitted the incorrect version of the revised edition. We are sincerely sorry for this. To rectify this, we have now included the comparisons on an anonymous webpage at https://anonymous-freemorph.github.io/. If feasible, we kindly invite the reviewer to explore our anonymous website for a visual representation of the comparisons.

---

### Author Response · Authors · 2024-12-01

We thank all reviewers for their time and effort in reviewing our paper!

Below, we summarize the changes made according to the reviews:

1. We compare our method with "Attention Interpolation for Text-to-Image Diffusion" and "Smooth Diffusion" to demonstrate our superiority (`#XAhp`).

2. We discuss more about Eq. (6) for slerp (`#XAhp`).

3. We show the difference between DCT and FFT and illustrate their influence on the generated image transitions (`#XAhp`).

4. We update the decimal digits in the tables and rectify the writing typos (`#XAhp`, `#NcSG`).

5. We discuss the nonlinearity of the multi-step denoising process (`#JPFT`).

6. We clarify what is a good image morphing (`#JPFT`).

7. We conduct user studies to evaluate different methods in a subjective manner (`#JPFT`, `#NcSG`, `#qmRm`).

8. We provide further explanation and rationale behind the design of our proposed modules, offering a detailed justification (`#JPFT`).

9. We explain the positive influence of Eq. (5) on identity preservation (`#JPFT`).

10. We clarify the reason for the name of our modules. We also modify their names for more accuracy (`#JPFT`).

11. We improve the caption for Fig.2 and Fig.3 (`#NcSG`).

12. We clarify the components of our introduced benchmark and improve the writing (`#NcSG`).

13. We clarify the process of reverse diffusion steps and forward denoising steps (`#NcSG`).

14. We clarify the design of our experiments "Ours (Var-A)" (`#NcSG`).

15. We illustrate the importance of smooth transition and identity preservation for image morphing (`#qmRm`).

16. We clarify the difference between smooth transition and directional transition (`#qmRm`).

We sincerely thank all reviewers and the AC(s) again for their valuable suggestions, which have greatly helped strengthen our paper.

If you have any further questions, we would be happy to discuss them!

---

### Meta-Review · Area_Chair_oWy7 · 2024-12-23

**Metareview:**

This paper proposes a tuning-free method for image morphing that accommodates significantly different semantics and image layouts. The authors introduce two key techniques—guidance-aware spherical interpolation and step-oriented motion flow—that yield visually pleasing transitions between source and target images. However, in addition to multiple spelling and typesetting issues, the reviewers identified several technical concerns, particularly regarding comparisons with closely related research (e.g., T2I Diffusion and Smooth Diffusion), subjective evaluations, and the rationale behind key components. The authors provided feedbacks, yet three of them remained somewhat hesitant to recommend acceptance. The Area Chair also notes that external links were provided for some experimental results, but the risk revealing reviewers' identities may exist. Although the fourth reviewer’s assessment (Reviewer qmRm) was more positive, the Area Chair observed discrepancies between the review’s tone and the final rating, reducing its overall weight in the decision. Taking all feedback into account, especially the reservations of the first three reviewers, the Area Chair has decided to reject the manuscript. The authors are encouraged to resubmit to a future venue, including all necessary supportive materials within the official review package rather than relying on external websites.

**Additional Comments On Reviewer Discussion:**

During the review process, Reviewer XAhp raised concerns about the potential risk of identity disclosure when accessing third-party links. This concern could not be independently verified by the reviewers or ACs. Subsequently, the authors provided an alternative link with additional experimental results. However, the perceived risk may have discouraged reviewers from exploring these materials, which limited the authors' opportunity to address critical concerns and potentially sway reviewer opinions.

The AC strongly advises the authors to include all supportive materials directly within the review package in their next submission. Furthermore, Reviewer qmRm was unable to respond to the AC's request to double-check the review report and score. Consequently, the AC relied more heavily on the assessments of the first three reviewers when making the final recommendation.

---

### Decision · Program_Chairs · 2025-01-22

Reject